Journal of Data-centric Machine Learning Research (2026)   Submitted 11/24; Revised 06/25; Published 03/26

# LOCKED: A Dataset of Sociodemographic, Economic, Health and Living Features to Assess Mental Health Impact of the Spanish Lockdown during COVID-19

**Alberto Nogales**                  ALBERTO.NOGALES@CEIEC.ES
*CEIEC Research Institute*
*Universidad Francisco de Vitoria*
*Pozuelo de Alarcón (Madrid), Carretera Pozuelo-Majadahonda km. 1,800. 28223, Spain*

**Alfredo Guitian**                  9005692@ALUMNOS.UFV.ES
*CEIEC Research Institute*
*Universidad Francisco de Vitoria*
*Pozuelo de Alarcón (Madrid), Carretera Pozuelo-Majadahonda km. 1,800. 28223, Spain*

**Blanca Mellor-Marsá**               BMELLORMARSA@GMAIL.COM
*Independent researcher*

**Álvaro J. García-Tejedor**             A.GTEJEDOR@CEIEC.ES
*CEIEC Research Institute*
*Universidad Francisco de Vitoria*
*Pozuelo de Alarcón (Madrid), Carretera Pozuelo-Majadahonda km. 1,800. 28223, Spain*

**Reviewed on OpenReview:** *https://openreview.net/forum?id=fQYcDj8fNG&noteId=KMST4n2LSa*

**Editor:** Fernando Perez-Cruz

## Abstract

The COVID-19 pandemic, which began in late 2019 in Wuhan, China, quickly escalated into a global crisis that affected nearly every aspect of life. Governments around the world implemented stringent public health measures to control the spread, including quarantines, social distancing, and lockdowns. In Spain, where the first cases emerged in January 2020, a nationwide lockdown was imposed on 14 March after the number of infections exceeded 5,000. Although these interventions were crucial for public health, they also presented significant social challenges, particularly for vulnerable groups. The primary goal of this study is to describe a dataset that combines psychological assessments of nine mental health conditions with sociodemographic, economic, living, and general health features. The data collected could potentially be used to identify the key factors that influenced mental health outcomes during lockdown. By analysing these data, the research seeks to shed light on the broader psychological effects of the pandemic and the factors that can exacerbate or mitigate these impacts. As an added value and to demonstrate the quality and potential of the dataset for mental health research, baseline machine learning models were developed, achieving performance metrics that exceed 80% in the case of the diagnostic classifiers. LOCKED is publicly available at https://zenodo.org/records/17179227.

**Keywords:** COVID-19, Mental health, Lockdown, SA-45 test.

## 1. Introduction

In 2020, the COVID-19 pandemic, caused by the new SARS-CoV-2 virus, spread rapidly throughout the world after its discovery in Wuhan, China. Far more contagious than the previous SARS outbreak, it led to more than 300,000 cases and 13,000 deaths by March 2020, Radovanovic et al. (2020). The virus overwhelmed healthcare systems, caused shortages of medical supplies, and disrupted global economies through lockdowns and business closures. Efforts to control the virus included social distancing, mask-wearing, and the accelerated development of vaccines, reshaping healthcare and economies globally.

According to Grant et al. (2020), common symptoms of COVID-19 include fever, fatigue, cough, and shortness of breath. Rapid human-to-human transmission of the virus was well documented at the beginning of the pandemic Ahmed et al. (2018). In the absence of a vaccine or effective treatments, self-isolation and quarantine became essential strategies to contain the virus, Livingston et al. (2020). Governments around the world implemented strict lockdowns, such as the 51-day lockdown in Spain that began on 14 March 2020, Shah et al. (2020). These measures, combined with prolonged isolation, had a significant impact on mental health, leading to increased anxiety and depression, Brooks et al. (2020). Individuals lacking social or psychological support were especially vulnerable to distress regarding Rodríguez-Rey et al. (2020), while the socioeconomic consequences worsened the mental health crisis in vulnerable populations.

The COVID-19 pandemic prompted the enforcement of quarantine measures in many countries. Although these measures were essential to control the spread of the virus, they negatively affected the psychological well-being of people. The situation involved separation from loved ones, inability to engage in normal activities, fear of infection, and a loss of personal freedom, Brooks et al. (2020). The psychological toll was especially severe in prolonged cases, as seen in Spain, where lifestyle restrictions were imposed due to insufficient awareness and anticipation of the virus's rapid spread, Gismero-González et al. (2020).

Research shows that enforced isolation can have a significant impact on many aspects of life, often leading to psychological stress and, in some cases, mental health issues. In Spain, data from the first wave of the pandemic revealed that a significant portion of the adult population experienced symptoms of depression and anxiety, Rodríguez-Rey et al. (2020) & González-Sanguino et al. (2020). Older individuals, particularly those over 60, were notably affected, developing depressive symptoms and avoidant coping styles. Risk factors for poorer mental health included being female, having a history of mental illness, direct exposure to COVID-19, experiencing virus-related symptoms, or having an infected close relative, Miranda-Mendizabal et al. (2022).

The primary motivation of this work is to present a dataset that explores how various personal factors are linked to the mental health of the population during the COVID-19 lockdown in Spain. The main contribution of this work is to make this dataset freely available to the scientific community, integrating psychological assessments, such as the SA-45, with detailed personal, economic, living, and health information. By analysing this rich combination of data, researchers can examine how diverse socioeconomic and personal factors affected mental health during the lockdown. As a complementary contribution, we demonstrate the utility and quality of the dataset by evaluating its performance in various machine learning (ML) models. We consider this dataset to be entirely novel because, to

the best of our knowledge, it is the only one that captures the impact of the COVID-19 lockdown on the mental health of the Spanish population, with data collected while the situation was still unfolding.

Our dataset aims to address literature gaps, particularly in terms of the role of sociodemographic, living, economic, and health determinants in mental health outcomes. These factors, which include income level, employment status, living conditions, and general health, can profoundly influence an individual's ability to cope with stress, depression, or anxiety. By providing a comprehensive dataset that captures these dimensions, we offer a valuable resource for exploring the complex interactions between mental health and social conditions during the lockdown that took place in Spain due to COVID-19.

This paper details the methodology used to compile and validate the dataset, providing initial insights into the observed trends. The dataset enables the evaluation of psychological processes during the lockdown and facilitates the use of artificial intelligence models to predict mental health outcomes based on individual and societal factors. This resource will support future research to develop more targeted mental health interventions during crises or implement social benefits for vulnerable groups.

The rest of the paper is organised as follows. Section 2 compiles different works using or presenting similar datasets. Section 3 describes how the data was gathered and the methods used to demonstrate the usefulness of the dataset. Section 4 shows different results of the dataset in Machine Learning models. Finally, Section 5 concludes and proposes a few future works.

## 2. Related work

The impact of COVID-19 has led to the publication of numerous studies utilising datasets like the one described in this paper. These studies measure the influence of various features on mental health due to the disease and its consequences. Below, we review several studies that use such datasets.

Sadegh-Zadeh et al. (2022) uses the COVID Impact Survey dataset[1], which compiles demographic and social factors such as income, education, trust, and social connections. These factors were collected in April, May, and June 2020. The data was analysed using Random Forest (RF), Support Vector Machines (SVM), Naïve Bayes (NB), and Logistic Regression (LR) models, with the target variable based on the frequency of psychological issues (anxiety, depression, and physical reactions) experienced over 1, 3, or 5 days per week. Wang et al. (2021) examines the mental health of 5,108 Chinese medical workers[2] during the pandemic. They used 32 features, such as age, type of employment, duration of sleep, and intensity of work, to predict conditions such as anxiety and depression with a novel neural model. Samuelson et al. (2022) explores risk and resilience factors affecting mental health outcomes, including depression, anxiety, Post-Traumatic Stress Disorder (PTSD), and somatic symptoms, using a non-public dataset of 467 U.S. adults. Features include demographic, medical, COVID-related, and psychological resilience factors, with Machine Learning models like RF, XGBoost, and SVM. Rezapour and Hansen (2022) uses a public

---

1. https://data.world/associatedpress/covid-impact-survey-public-data
2. https://github.com/Hu-Li/mental-health-dataset

dataset[3] of 518 frontline healthcare workers, incorporating features such as demographics, professional roles, behavioural habits, and COVID-19 impact, to train models such as RF, Gradient Boosting (GB), and XGBoost. The target variable measures mood changes during the pandemic. Prout et al. (2020) analyses a non-public dataset on psychological distress among 2,787 participants during the COVID-19 pandemic, focusing on factors such as anxiety, depression, and PTSD. They used RF and Regression Tree models for this analysis. The dataset includes features such as demographics, childhood trauma, emotion regulation strategies, and somatisation. While these datasets measure the impact of COVID-19 on mental health, none of them focus on Spanish individuals under confinement, and some datasets are not publicly available.

Other studies have specifically examined mental health during the COVID-19 lockdown. Herbert et al. (2021) explores mental health, personality, and behavioural changes in university students from Germany and Egypt during the first lockdown in May 2020 (the data is not publicly available). The target variables in the Machine Learning models are the Big Five personality traits, analysed using Support Vector Regression (SVR) and Gradient Boosting Regression (GBR). Glowacz and Schmits (2020) investigates the impact of lockdown on mental health across different age groups (18–30, 30–50, and over 50) in French-speaking countries, primarily Belgium (dataset is available upon request). Features include living environment, professional status, social contact, proximity to contamination, alcohol use, and intolerance of uncertainty, with anxiety and depression serving as outcome variables in multiple regression models. Di Giuseppe et al. (2020) utilises a dataset of 5,683 Italians during the early COVID-19 lockdown in March 2020, which can be accessed upon request. Features included demographic information and COVID-19 exposure, and LR was applied to measure psychological outcomes such as distress, depression, and post-traumatic stress symptoms. Ntakolia et al. (2022) examines the psychological effects of lockdown on children and adolescents in Greece who have pre-existing mental health conditions. The dataset, which includes 71 heterogeneous features (e.g., demographics, family life, daily activities, mood states), is available upon request and was analysed using RF, XGBoost, SVM, and LR models, with mood changes as the target variable (improvement, stability, or deterioration). Flesia et al. (2020) investigates stress levels in 2,053 adults in a non-public dataset during the pandemic lockdown, identifying sociodemographic factors (age, gender, income) and psychological traits (coping strategies, emotional stability, self-control) as targets in ML models such as LR, SVM, NB, and RF. Although these studies focus on the impact of COVID-19 on mental health during different lockdowns, they do not address the highly restrictive lockdown in Spain.

Several studies are noteworthy regarding the mental health impact of COVID-19 on the Spanish population. Simjanoski et al. (2022) examines the relationship between lifestyle behaviours and mental health, particularly depression and anxiety, during the early stages of the pandemic. This study uses data from 22,562 participants in Spain and Brazil, available upon request, to train Elastic Net, RF, and XGBoost models. Tubío-Fungueiriño et al. (2022) focuses on Spanish patients with Obsessive-Compulsive Disorder (OCD), using ML models to predict Y-BOCS scores, self-perceived anxiety, depression, and suicidal thoughts. The same dataset is used by Alonso et al. (2021), complemented by data from 237 con-

---

3. https://www.openicpsr.org/openicpsr/project/127081/version/V1/view?path=/openicpsr/127081/fcr:
   versions/V1&type=project

trols, to train linear regression models. In both cases, the dataset is available upon request. Chulià et al. (2020) studies the mental health of 523 Spanish adolescents (ages 13–17), focusing on emotional symptoms such as depression, anxiety, and stress, measured using the DASS-21 scale and analysed using Multiple Logistic Regression. The dataset is only available by contacting the authors. Finally, Aguilar-Latorre et al. (2022). investigates how Social Determinants of Health (SDH) influenced depression during the strict seven-week lockdown in Zaragoza, Spain. The study considers features such as living conditions, access to green spaces, housing quality, social support, and access to healthcare. However, this study does not apply any ML models, and the dataset is not publicly available. Although these datasets involve Spanish populations, the data were not always collected during the lockdown.

As far as we know, the dataset described in this paper is completely novel due to the following features. Openness: Most of the datasets described in this section are only available upon request. Heterogeneity, individuals are restricted to the Spanish population that has filled in the questionnaires while being under lockdown. Considering both features, none of the datasets in this section accomplishes both.

## 3. LOCKED dataset

The dataset comprises information from 1,030 individuals, gathered through a 96-item questionnaire about their personal, living, economic, and health conditions before and during the COVID-19 quarantine, which was complemented with a psychological test. The questionnaire was designed by one of the authors, who has a PhD in psychology. Participants were recruited through convenience and snowball sampling via online means, in compliance with lockdown restrictions. The survey was distributed between May 2 and May 9, 2020, using Google Forms. On this date, some regions of Spain entered stage 0, which marked the beginning of the relaxation of strict lockdown measures, allowing people to walk during designated time slots. Invitations were shared on social media platforms (Twitter, LinkedIn, Facebook, Instagram), professional and particular WhatsApp groups, mailing lists of research communities, and among personal contacts. No incentives were offered, and participation was entirely voluntary. Due to lockdown constraints, the data was gathered without expert supervision; all participants consented to their information being used in the study.

Eligibility criteria included being at least 18 years old and residing in Spain during the national COVID-19 lockdown. While this online strategy enabled rapid data collection during a critical phase of the pandemic, it inherently limits the sample's representativeness. In particular, it likely underrepresents older adults, individuals with limited internet access, and less digitally connected populations. These factors may introduce sampling bias, which in turn constrains the external validity and generalizability of the findings. Users of the dataset should take these limitations into account when interpreting the results or applying models trained on this data.

The questionnaire had two main components: the Ad hoc Sociodemographic Survey and the Symptom Assessment-45 Questionnaire (SA-45), described in Holgado-Tello et al. (2019) & Sandín et al. (2008). The first section consisted of 51 questions designed to capture participants' personal, living, economic, and health conditions (see the first Appendix, Personal

questionnaire). The second part was the SA-45, the Spanish version of a tool designed to assess general psychopathology, Davison et al. (1997). This self-administered questionnaire contains 45 items (see the second Appendix, SA-45 Test) and measures nine symptom dimensions: obsession-compulsion, interpersonal sensitivity, hostility, anxiety, somatisation, paranoid ideation, phobic anxiety, psychoticism, and depression. Each condition contains five items, with responses rated on a Likert scale from 0 ("Not at all") to 4 ("Very much or extremely"). The total scores range from 0 to 180, while the dimension scores range from 0 to 20, with higher scores indicating a higher level of psychopathological symptoms.

### 3.1. Feature curation and preprocessing stage

Data cleaning involved handling poorly collected, highly variable features, outliers, and missing values using various strategies. Instances with critical formatting errors were also removed, resulting in a final dataset of 981 observations.

In terms of the variables, open-ended fields with high variability, such as place of birth, current postal code, and specific job titles, were excluded due to their broad nature and the excessive noise they introduced into the analysis. In total, 10 features were discarded due to this condition.

In the cases where empty values were found in cells, the following strategies were implemented. In 51 cases where features were continuous, the value NaN was entered because the absence of a value could potentially be interpreted as 0 by the individual (e.g., having children). Meanwhile, there were 43 categorical cells with null values that were also filled with NaN. Using NaNs allows future users of the dataset to decide how to handle these cases. To train the ML models presented in this work, we have left 0 values for NaNs in continuous features and the most frequent value for categorical ones.

In one of the cell columns, an outlier was detected and replaced with the median of the established values. This value was considered an outlier because it corresponded to the "Minors in your care" feature, with a value of 21.

In terms of data heterogeneity, we have provided the distribution and entropy of the different categorical features and the average and standard deviation values for the continuous ones, which can be found in Appendices 3 (Distribution of Categorical Variables) and 4 (Distribution of Numerical Variables). Apart from that, we have analysed in detail the postcode feature to ensure that individuals from different parts of Spain are represented in the dataset. In this way, of the 17 regions and 2 autonomous cities in Spain, 18 are represented, with a minimum of one individual (Ceuta) and a maximum of 370 individuals (Community of Madrid). This indicates a wide geographical distribution of responses.

### 3.2. Data labelling

Initially, it was necessary to define the classes of the target feature to enable the application of ML models to classify individuals based on the collected features. The primary objective was to label individuals according to psychological processes. To achieve this, we used the SA-45 test, which follows clinical methodologies commonly used in practice. The SA-45 test comprises 45 items, each of which is assigned to one of the nine dimensions of psychological distress. The sum of the scores for the items in each scale yields a scale-specific score.

- Hostility (items 7, 34, 35, 39, 43): This dimension reflects anger, irritability, and aggression. A high score may suggest a tendency to react with anger or experience feelings of hostility towards others.

- Somatisation (items 18, 23, 26, 29, 31): This refers to experiencing physical symptoms without an apparent medical cause, which are often linked to emotional or psychological factors. A high score may suggest a tendency to report multiple physical complaints.

- Depression (items 9, 10, 11, 27, 42): This scale measures symptoms of depression, including profound sadness, loss of interest, fatigue, and hopelessness. A high score suggests a significant presence of depressive symptoms.

- Obsessive-Compulsive (items 16, 20, 21, 25, 28): This dimension relates to the presence of intrusive thoughts (obsessions) or ritualistic behaviours (compulsions). A high score indicates a greater tendency towards obsessive-compulsive patterns.

- Anxiety (items 6, 12, 30, 38, 41): This scale assesses symptoms of anxiety, such as excessive worry, tension, and fear. A high score reflects a high presence of anxiety symptoms.

- Interpersonal Sensitivity (items 14, 15, 17, 32, 36): This dimension evaluates feelings of inferiority and self-criticism in social interactions. A high score indicates heightened sensitivity to criticism and feelings of inadequacy in social situations.

- Agoraphobia (items 3, 8, 22, 24, 37): This refers to the fear of being in situations where it might be difficult to escape or where help might be unavailable, particularly in the case of a panic attack. A high score suggests a stronger presence of these fears.

- Paranoid Ideation (items 2, 5, 19, 40, 44): This scale measures paranoid thoughts, such as excessive distrust or the belief that one is being persecuted. A high score indicates a significant tendency towards paranoia.

- Psychoticism (items 1, 4, 13, 33, 45): This dimension assesses symptoms associated with psychotic disorders, such as hallucinations, unusual thoughts, and strange behaviour. A high score indicates a greater presence of psychotic symptoms.

Currently, there is no normative data available to help interpret the SA-45 scores. However, elevated scores can be defined as values exceeding the mean plus one standard deviation. This criterion is consistent with the approach proposed by Dang et al. (2021) for the SCL-90, from which the SA-45 is derived as its abbreviated form. Then, the score for each of the nine scales can be calculated by dividing the total score by the number of items in the scale.

### 3.3 Studied features

The remaining variables were encoded for use as input in classifiers. The dataset comprises 41 variables, including 20 multicategorical and 7 binary. To enhance model performance, the original variables were grouped into the following categories while maintaining logical

separation. Efforts were made to balance the number of features across categories.

Demographic Characteristics (11 features): This category includes participants' gender, age, marital status, place of birth, documentation status, nationality, educational attainment, and family structure.

Living Environment (10 features): Variables in this category describe the type of housing, housing characteristics, number of cohabitants, and the frequency of social interactions before and after the quarantine period.

Economic Status (11 features): This category encompasses employment status before and after quarantine, job type, working hours before and after quarantine, income levels across the same periods, and the capacity to manage monthly expenses and debts.

Health Impact (9 features): This category includes information on general health conditions and the specific impacts of COVID-19, including personal health outcomes and caregiving responsibilities during the pandemic.

By applying this approach, our dataset is split into four separate datasets, which compile similar features for a more focused analysis (see the fifth Appendix).

## 4. Classification and Predictive Benchmarking (Illustrative Use Cases)

As an added value of the dataset, two main experiments have been conducted to demonstrate that the compiled data is sufficient to achieve good performance metrics across different models. For the first set of experiments, since the dataset was collected to diagnose nine psychological conditions using the SA-45 test, it is appropriate to evaluate its performance in classification models. In this case, we established a benchmark using six well-known supervised ML models. For the second set of experiments, we developed a series of multicategorical regression models to predict the aggregated SA-45 score for an individual. The different models in this section are formalised below. First, the used classifiers are described.

Decision Trees (DT) employ a "divide and conquer" approach, splitting data into nodes based on features until a class is assigned at the leaves, Breiman (2017). RF, an ensemble method, constructs multiple decision trees on random data samples and subsets of features, combining their outputs to enhance accuracy and reduce overfitting, Breiman (2001). SVM, defined by Cortes (1995), aims to maximise the margin between classes by identifying the optimal hyperplane in high-dimensional space. GB, introduced by Friedman (2002), sequentially builds a strong model by combining weaker ones, typically decision trees, with each new model correcting errors from the previous one to minimise loss. NB assumes that the features are conditionally independent given the target class and calculates the overall probability as the product of the individual probabilities, Mitchell (2007). LR, a linear classification model, predicts the probability of a class using a logistic function. It can also handle non-linear data through polynomials or interaction terms, McCullagh (2019). Finally, the Multilayer Perceptron (MLP), described in Rumelhart et al. (1986), is a neural network with multiple layers of neurons, where each connection is weighted and adjusted during training to minimise prediction errors.

As the second group of experiments is based on regression models, we first formalise this technique. In this case, a separate regression model is created for each subset, each comprising different features. Since these models aim to predict the total SA-45 score for

an individual, they are implemented as multiple linear regression models. This is defined in Ganesh (2010) as a statistical method used to model the relationship between a single response variable and two or more predictor (regressor) variables.

In the classifier experiments, for each of the four subsets, we trained one of the aforementioned models and applied the following good practices, Nogales et al. (2024). First, the datasets were partitioned into training, validation, and test sets. The training and validation sets were used to optimise the model's hyperparameters, and the test set was reserved for evaluating the final model's performance. Second, a grid search strategy was applied to find the best hyperparameter configuration. Table 1 shows hyperparameters and values used to fine-tune each model.

**Table 1.** ML models' hyperparameters for grid search

| ML model | Hyperparameters |
|---|---|
| DT | Maximum depth: [None, 10, 20, 30] |
| | Minimum samples split: [2, 5, 10] |
| | Minimum samples per leaf: [1, 2, 4] |
| | Split Criterion: ['Gini', 'Entropy'] |
| RF | Number of estimators: [50, 100, 150] |
| | Maximum depth: [None, 10, 20] |
| | Minimum samples split: [2, 5] |
| | Minimum samples per leaf: [1, 2] |
| SVM | Regularization parameter (C): [0.1, 1, 10] |
| | Kernel: ['Linear', 'Radial Basis Function (RBF)', 'Polynomial'] |
| | Degree (for Polynomial kernel): [3, 4, 5] |
| | Gamma: ['Scale', 'Auto'] |
| GB | Number of estimators: [50, 100, 150] |
| | Learning rate: [0.01, 0.1, 0.2] |
| | Maximum depth: [3, 4, 5] |
| | Subsampling ratio: [0.8, 1.0] |
| NB | Variance smoothing: [1.0 . . . 1e-9] |
| | Alpha: [0.1, 0.5, 1.0, 1.5, 2.0, 5.0] |
| | Class prior estimation: [True, False] |
| LR | Regularization parameter (C): [0.01, 0.1, 1, 10, 100] |
| | Regularization type: [L1, L2] |
| | Optimization algorithm: ['Library for Large Linear Classification', 'Limited-memory Broyden–Fletcher–Goldfarb–Shanno (LBFGS)'] |
| | Maximum iterations: [100, 200, 500] |
| MLP | Hidden layer sizes: [(50,), (100,), (50, 50)] |
| | Activation function: ['ReLU', 'tanh'] |
| | Solver: ['Adam', 'Stochastic Gradient Descent (SGD)'] |
| | Alpha: [0.0001, 0.001] |
| | Learning rate type: ['Constant', 'Adaptive'] |

An 80-20% split was applied to divide the dataset into training and test subsets, with randomised instance allocation to minimise bias. As the majority class comprises people who are not experiencing a psychological condition, an undersampling strategy was applied to stratify the number of instances in each class for the different classifiers. This information can be found in Table 2.

**Table 2.** Train and split sets for each classifier

| ML model | Training set | Test set |
|---|---|---|
| Hostility | 233 | 59 |
| somatization | 252 | 64 |
| Depression | 252 | 64 |
| Obsession-Compulsion | 267 | 67 |
| Anxiety | 270 | 68 |
| Interpersonal Sensitivity | 270 | 68 |
| Agoraphobia | 198 | 50 |
| Paranoid Ideation | 208 | 52 |
| Psychoticism | 224 | 56 |

Hyperparameter optimisation was conducted using a grid search strategy, which systematically explored multiple combinations of parameters Bergstra et al. (2011). To enhance the robustness of the evaluation, k-fold cross-validation was employed to allow a more comprehensive assessment of model performance. Since certain ML models initialise hyperparameters randomly, this, combined with randomised data partitioning, may positively impact model performance. Model performance was evaluated using the accuracy metric, defined as the proportion of correctly predicted instances.

The outcomes of this approach are presented in Table 3, which summarises the accuracy of the training, validation, and test sets across four subsets for each of the eight classifiers. The results are reported as the mean and standard deviation, reflecting the variability introduced by k-fold cross-validation. The training and evaluation code is available at https://github.com/ufvceiec/LOCKED. The Table also identifies the best-performing model for each case, alongside the corresponding accuracy values.

The training of the ML models was conducted on a desktop computer equipped with an Intel(R) Core(TM) i5-4590 CPU running at 3.30 GHz. The system features 16 GB of RAM (15.9 GB usable). In total, 196 models (combinations of subsets, classes, and ML models) plus their grid search were trained, resulting in a total duration of 1 hour, 3 minutes, and 23 seconds.

**Table 3.** Accuracy applying benchmarks.

| Training subset | Binary classifier | Best ML model | Train | Validation | Test |
|---|---|---|---|---|---|
| | Hostility | DT | 91.15% ± 2.50 | 85.16% ± 1.48 | 81.36% |

Demographic characteristics

| | | | | | |
|---|---|---|---|---|---|
| | somatization | SVM | 81.70% ± 5.86 | 76.36% ± 3.09 | 87.50% |
| | Depression | SVM | 85.09% ± 4.47 | 80.27% ± 2.85 | 87.50% |
| | Obsession-Compulsion | RF | 90.96% ± 2.40 | 82.40% ± 1.50 | 89.55% |
| | Anxiety | SVM | 81.89% ± 5.42 | 77.81% ± 3.47 | 89.71% |
| | Interpersonal Sensitivity | DT | 89.60% ± 2.42 | 87.59% ± 1.01 | 85.29% |
| | Agoraphobia | RF | 92.76% ± 1.25 | 85.89% ± 0.53 | 90.00% |
| | Paranoid Ideation | DT | 92.23% ± 1.69 | 84.52% ± 1.10 | 92.31% |
| | Psychoticism | MLP | 86.83% ± 7.37 | 79.95% ± 3.07 | 87.50% |
| | Hostility | SMV | 83.94% ± 10.16 | 76.75% ± 6.93 | 88.14% |
| | somatization | SVM | 84.14% ± 9.70 | 76.85% ± 7.98 | 87.50% |
| | Depression | MLP | 73.34% ± 3.67 | 71.04% ± 3.46 | 84.38% |
| Living environment | Obsession-Compulsion | RF | 95.35% ± 2.12 | 88.36% ± 0.78 | 89.55% |
| | Anxiety | SVM | 82.2% ± 10.67 | 75.35% ± 7.49 | 94.12% |
| | Interpersonal Sensitivity | GB | 94.44% ± 5.40 | 80.75% ± 1.74 | 83.82% |
| | Agoraphobia | MLP | 90.60% ± 7.83 | 81.22% ± 3.77 | 90.00% |
| | Paranoid Ideation | RF | 96.09% ± 1.98 | 85.63% ± 0.67 | 90.38% |
| | Psychoticism | RF | 96.50% ± 1.63 | 79.53% ± 1.35 | 92.86% |
| | Hostility | SVM | 89.22% ± 7.89 | 83.76% ± 5.56 | 91.52% |
| | somatization | RF | 96.30% ± 1.30 | 87.86% ± 0.91 | 87.50% |
| | Depression | DT | 93.12% ± 2.29 | 78.73% ± 1.25 | 85.29% |
| Economic status | Obsession-Compulsion | RF | 97.64% ± 0.98 | 88.20% ± 0.71 | 88.06% |
| | Anxiety | RF | 97.17% ± 1.12 | 85.75% ± 0.51 | 85.29% |
| | Interpersonal Sensitivity | DT | 91.98% ± 3.01 | 81.55% ± 1.48 | 91.18% |
| | Agoraphobia | SVM | 87.94% ± 9.88 | 80.12% ± 6.78 | 94.00% |
| | Paranoid Ideation | RF | 95.66% ± 1.56 | 85.40% ± 1.37 | 96.15% |
| | Psychoticism | NB | 83.92% ± 4.15 | 82.60% ± 3.74 | 87.50% |
| | Hostility | DT | 95.74% ± 1.15 | 93.12% ± 0.35 | 96.61% |
| | somatization | RF | 96.70% ± 0.15 | 96.15% ± 0.68 | 92.19% |
| | Depression | GB | 96.84% ± 0.79 | 95.37% ± 0.78 | 95.31% |
| Health impacts | Obsession-Compulsion | DT | 94.01% ± 0.61 | 93.51% ± 0.70 | 98.51% |
| | Anxiety | RF | 95.40% ± 0.16 | 95.31% ± 0.41 | 92.65% |
| | Interpersonal Sensitivity | DT | 95.68% ± 0.63 | 94.69% ± 0.70 | 92.65% |

| | | | | |
|---|---|---|---|---|
| Agoraphobia | NB | 92.79% ± 8.67 | 92.61% ± 8.91 | 98.00% |
| Paranoid Ideation | DT | 92.30% ± 0.68 | 92.00% ± 1.13 | 100% |
| Psychoticism | SVM | 87.33% ± 7.45 | 85.89% ± 7.39 | 91.07% |

A fundamental objective when training ML models is to avoid overfitting and underfitting. This can be achieved by addressing the bias-variance trade-off Belkin et al. (2019), which involves balancing model complexity with its ability to generalise to unseen data. This trade-off is crucial for optimising both performance and robustness in Machine Learning. Bias represents the model's ability to capture the underlying patterns in the data, while variance reflects its sensitivity to minor variations during training. The performance (bias) of all the ML models evaluated was satisfactory, with accuracy values exceeding 80% and many models achieving accuracies above 90%. Variance was considered acceptable if it did not exceed a difference of 10 percentage points.

Although accuracy is a widely used metric for evaluating model performance, more comprehensive metrics, such as sensitivity and specificity, provide deeper insights into model behaviour by accounting for False Positives (FP) and False Negatives (FN). Sensitivity quantifies the proportion of actual positives that are correctly identified, while specificity measures the proportion of actual negatives that are correctly identified. These metrics are crucial for understanding the nature of the errors made by the model, particularly in scenarios where FP or FN have significant implications. Tables 4 and 5 present analogous sensitivity and specificity results to those in Table 3. Since the best-performing model remains consistent across these evaluations, the corresponding column has been omitted from these tables.

**Table 4.** Sensitivity applying benchmarks.

| Training subset | Binary classifier | Train | Validation | Test |
|---|---|---|---|---|
| Demographic characteristics | Hostility | 88.61% ± 0.19 | 81.76% ± 0.84 | 65.52% |
| | Somatization | 78.19% ± 0.09 | 72.03% ± 1.01 | 81.25% |
| | Depression | 97.81% ± 0.02 | 94.09% ± 0.14 | 81.25% |
| | Obsession-Compulsion | 83.02% ± 0.06 | 75.55% ± 0.79 | 81.82% |
| | Anxiety | 76.77% ± 0.05 | 98.10% ± 0.04 | 79.41% |
| | Interpersonal Sensitivity | 83.40% ± 0.16 | 67.63% ± 0.22 | 73.53% |
| | Agoraphobia | 91.67% ± 0.01 | 96.41% ± 0.02 | 90.00% |
| | Paranoid Ideation | 89.41% ± 0.05 | 78.61% ± 1.77 | 96.15% |
| | Psychoticism | 96.26% ± 0.05 | 87.44% ± 0.53 | 82.14% |
| Living environment | Hostility | 82.28% ± 0.04 | 78.52% ± 1.22 | 79.31% |
| | Somatization | 83.13% ± 0.03 | 77.86% ± 0.43 | 81.25% |
| | Depression | 74.38% ± 2.72 | 69.23% ± 2.71 | 84.38% |
| | Obsession-Compulsion | 88.25% ± 0.05 | 81.33% ± 0.54 | 87.88% |
| | Anxiety | 83.25% ± 0.04 | 76.32% ± 0.56 | 91.18% |

| | Binary classifier | Train | Validation | Test |
|---|---|---|---|---|
| | Interpersonal Sensitivity | 88.26% ± 0.02 | 75.43% ± 0.78 | 81.25% |
| | Agoraphobia | 86.85% ± 0.06 | 78.68% ± 0.18 | 88.00% |
| | Paranoid Ideation | 95.13% ± 0.03 | 78.07% ± 0.24 | 92.31% |
| | Psychoticism | 94.37% ± 0.04 | 80.58% ± 0.21 | 89.29% |
| Economic status | Hostility | 90.60% ± 0.01 | 86.11% ± 1.20 | 86.21% |
| | Somatization | 93.43% ± 0.01 | 83.05% ± 0.29 | 84.38% |
| | Depression | 90.04% ± 0.04 | 71.44% ± 0.48 | 78.13% |
| | Obsession-Compulsion | 97.76% ± 0.00 | 88.09% ± 0.30 | 93.94% |
| | Anxiety | 96.32% ± 0.00 | 86.70% ± 0.25 | 82.35% |
| | Interpersonal Sensitivity | 88.31% ± 0.03 | 78.29% ± 0.06 | 88.24% |
| | Agoraphobia | 93.43% ± 0.01 | 90.21% ± 0.29 | 88.00% |
| | Paranoid Ideation | 89.66% ± 0.01 | 80.03% ± 0.58 | 92.31% |
| | Psychoticism | 76.33% ± 0.05 | 76.70% ± 0.40 | 75.00% |
| Health impacts | Hostility | 94.02% ± 0.00 | 91.05% ± 0.15 | 93.10% |
| | Somatization | 93.66% ± 0.01 | 93.71% ± 0.22 | 84.38% |
| | Depression | 94.44% ± 0.00 | 92.00% ± 0.29 | 90.63% |
| | Obsession-Compulsion | 90.29% ± 0.02 | 86.31% ± 0.65 | 96.97% |
| | Anxiety | 91.87% ± 0.00 | 91.94% ± 0.06 | 91.18% |
| | Interpersonal Sensitivity | 93.28% ± 0.02 | 90.19% ± 0.28 | 88.24% |
| | Agoraphobia | 92.88% ± 0.02 | 92.22% ± 0.22 | 96.00% |
| | Paranoid Ideation | 85.61% ± 0.03 | 86.18% ± 0.64 | 100% |
| | Psychoticism | 83.98% ± 0.08 | 83.92% ± 0.92 | 82.14% |

The above results allow us to assess whether the models encounter difficulties in correctly identifying healthy individuals, potentially misdiagnosing them as having a psychological condition. Such misdiagnoses could lead to unnecessary treatments, incurring economic and psychological consequences. Overall, most models achieve an accuracy level above 80%, with many exceeding 90%. However, some specific cases show lower performance. For the subset of demographic characteristics, challenges arise in diagnosing hostility, interpersonal sensitivity, and anxiety. In the living environment subset, hostility is difficult to diagnose. Finally, for the economic status subset, the models struggle with diagnosing depression and psychoticism. While the classifiers generally perform well, the demographic characteristics subset shows the greatest inconsistency values, particularly for the psychological condition of hostility. Notably, the health impacts subset demonstrates consistent performance across all conditions, with no significant issues observed.

**Table 5.** Specificity applying benchmarks.

| Training subset | Binary classifier | Train | Validation | Test |
|---|---|---|---|---|
| | Hostility | 94.35% ± 0.07 | 91.92% ± 0.50 | 96.67% |
| | Somatization | 99.19% ± 0.01 | 92.69% ± 0.20 | 93.75% |
| Demographic characteristics | | | | |

| | | | | |
|---|---|---|---|---|
| | Depression | 97.81% ± 0.02 | 94.09% ± 0.14 | 93.75% |
| | Obsession-Compulsion | 97.93% ± 0.01 | 94.09% ± 0.44 | 97.06% |
| | Anxiety | 98.10% ± 0.04 | 95.86% ± 0.08 | 100% |
| | Interpersonal Sensitivity | 95.56% ± 0.05 | 90.28% ± 0.45 | 97.06% |
| | Agoraphobia | 96.41% ± 0.02 | 90.09% ± 0.50 | 92.00% |
| | Paranoid Ideation | 97.82% ± 0.01 | 90.34% ± 0.50 | 88.46% |
| | Psychoticism | 96.89% ± 0.02 | 82.98% ± 0.76 | 92.86% |
| | Hostility | 100% ± 0.00 | 99.26% ± 0.02 | 96.67% |
| | Somatization | 99.41% ± 0.00 | 97.46% ± 0.10 | 93.75% |
| | Depression | 88.41% ± 1.24 | 81.01% ± 3.88 | 84.38% |
| | Obsession-Compulsion | 98.30% ± 0.01 | 92.49% ± 0.17 | 91.18% |
| Living environment | Anxiety | 98.68% ± 0.01 | 95.54% ± 0.15 | 97.06% |
| | Interpersonal Sensitivity | 98.11% ± 0.02 | 85.69% ± 0.59 | 94.12% |
| | Agoraphobia | 94.96% ± 0.02 | 88.03% ± 0.84 | 92.00% |
| | Paranoid Ideation | 100% ± 0.00 | 87.03% ± 0.63 | 88.46% |
| | Psychoticism | 98.00% ± 0.03 | 83.93% ± 0.58 | 96.43% |
| | Hostility | 100% ± 0.00 | 99.00% ± 0.04 | 96.67% |
| | Somatization | 98.82% ± 0.01 | 86.01% ± 0.56 | 93.75% |
| | Depression | 98.41% ± 0.01 | 86.70% ± 0.02 | 93.75% |
| | Obsession-Compulsion | 98.13% ± 0.01 | 89.41% ± 0.09 | 82.35% |
| Economic status | Anxiety | 99.28% ± 0.00 | 87.16% ± 0.35 | 88.24% |
| | Interpersonal Sensitivity | 94.39% ± 0.05 | 85.42% ± 0.60 | 94.12% |
| | Agoraphobia | 100% ± 0.00 | 96.67% ± 0.44 | 100% |
| | Paranoid Ideation | 100% ± 0.00 | 91.89% ± 0.37 | 100% |
| | Psychoticism | 93.76% ± 0.01 | 93.87% ± 0.09 | 100% |
| | Hostility | 100% ± 0.00 | 100% ± 0.00 | 100% |
| | Somatization | 100% ± 0.00 | 100% ± 0.00 | 100% |
| | Depression | 100% ± 0.00 | 100% ± 0.00 | 100% |
| Health impacts | Obsession-Compulsion | 99.25% ± 0.00 | 99.29% ± 0.02 | 100% |
| | Anxiety | 99.26% ± 0.00 | 99.33% ± 0.02 | 94.12% |
| | Interpersonal Sensitivity | 99.26% ± 0.00 | 99.33% ± 0.02 | 97.06% |
| | Agoraphobia | 100% ± 0.00 | 100% ± 0.00 | 100% |
| | Paranoid Ideation | 100% ± 0.00 | 100% ± 0.00 | 100% |
| | Psychoticism | 100% ± 0.00 | 100% ± 0.00 | 100% |

The metrics presented in Table 5 are particularly significant due to their direct implications for mental health. Low values suggest that individuals suffering from a psychological condition may be misdiagnosed as healthy, which can have serious consequences for their well-being and access to appropriate care. Analysing the subsets reveals that the health impacts subset demonstrates the best performance, aligning with the observed sensitivity

values. In the remaining cases, the results are satisfactory, with no values below 82% and many above 90%.

Although the models achieved good accuracy, sensitivity, and specificity, these measures only reflect classification ability and do not guarantee that predicted probabilities correspond to actual risk. Thus, calibration curves are essential to complement traditional metrics, ensuring that probability estimates are reliable and clinically meaningful. Calibration curves assess whether the predicted probabilities generated by a binary classifier accurately reflect the true likelihood of the outcome, Wilks (1990). In Fig. 1, we compile the calibration curves for the four best-performing models in the diagnosis of agoraphobia: RF, MLP, SVM and NB.

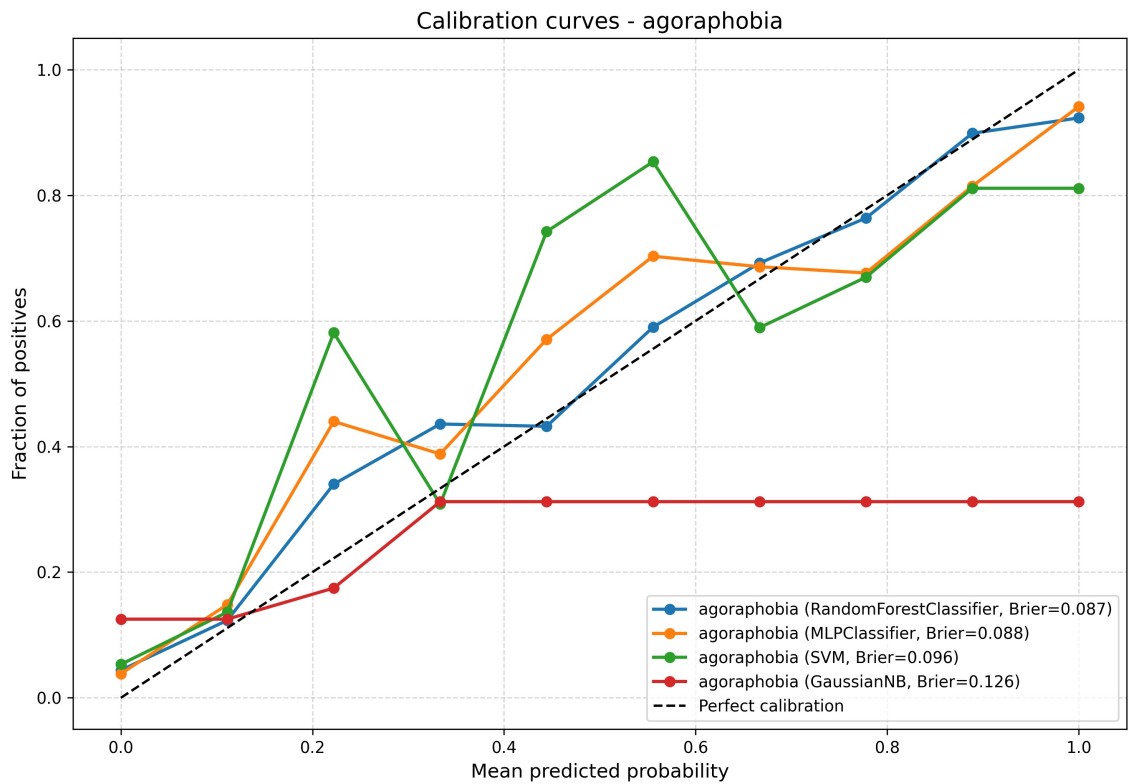

Figure 1: Calibration curves for best models diagnosing agoraphobia.

From the calibration curves for agoraphobia in Fig. 1, we can conclude the following. Random Forest (accuracy = 90%, Brier = 0.087) shows the most stable calibration. Predictions are close to the diagonal, with only mild underestimation between 0.3 and 0.4. Its distribution shows consistent and reliable estimates. This balance of good accuracy and excellent calibration makes it one of the most clinically reliable models.

MLP (accuracy = 90%, Brier = 0.088) performs similarly to Random Forest. Its calibration curve follows the diagonal well, with slight overestimation between 0.2 and 0.6, but remains close to good calibration overall. With similar accuracy and almost identical Brier score,

MLP is also a strong candidate for generating trustworthy probability estimates.

SVM (accuracy = 94%, Brier = 0.096) achieves higher accuracy than RF and MLP, but its calibration curve is unstable, oscillating between over- (0.2 and 0.4 to 0.5) and underestimation (from 0.6) across probabilities. This instability reduces the reliability of its probabilities, even though its classification accuracy is strong.

Gaussian Naive Bayes (accuracy = 98%, Brier = 0.126) attains the highest accuracy but exhibits the weakest calibration. Its curve flattens around 0.3, failing to represent higher levels of risk and producing systematically underconfident predictions. This mismatch between excellent classification accuracy and poor calibration highlights that NB's outputs, while discriminative, are not reliable as probability estimates.

Summarising, RF and MLP, with slightly lower accuracies (90%), provide well-calibrated probabilities that clinicians can trust when setting thresholds for intervention. Miscalibration can result in overdiagnosis (overestimation, leading to unnecessary interventions) or underdiagnosis (underestimation, leading to missed treatment opportunities). Therefore, although RF and MLP appear less accurate at first glance, their superior calibration makes them more appropriate for clinical decision-making in agoraphobia risk assessment.

Another way to complement the evaluation of the metrics is through permutation importance analysis, which reveals how each domain contributes distinct predictive factors. For this purpose, we selected the best-performing models within each subset, SVM, DT, and RF, for the prediction of paranoid ideation and anxiety and plotted Fig. 2.

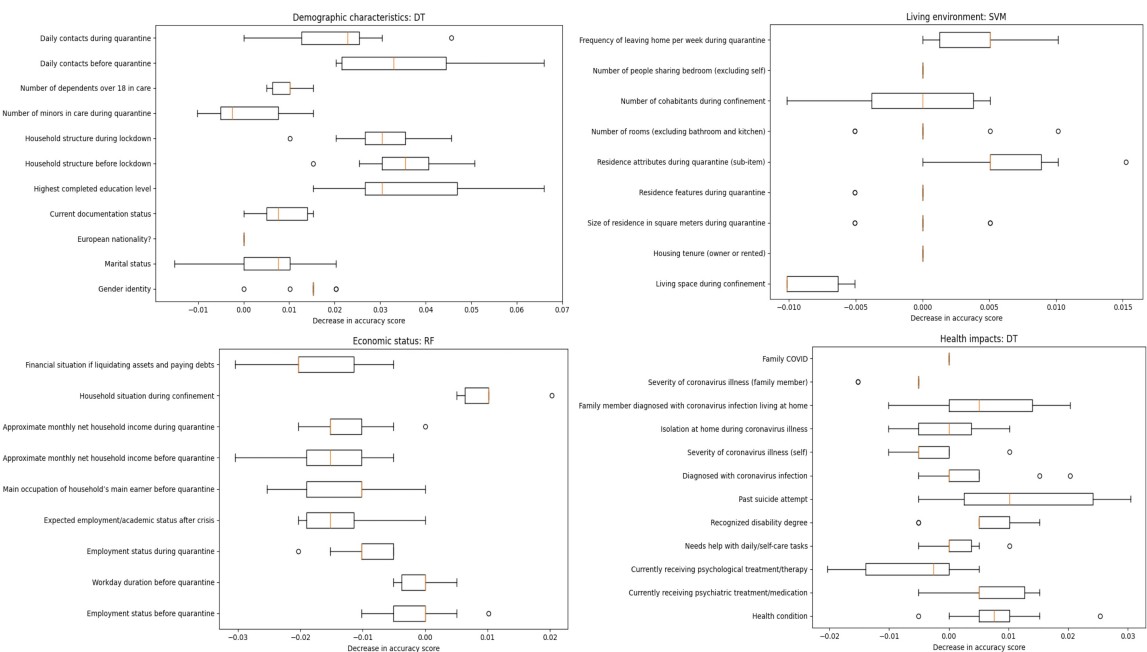

Figure 2: Permutation importance for best models in the four subsets.

Permutation importance from the previous Figure analyses highlighted that the demographic model for paranoid ideation was primarily driven by daily social contacts before and during quarantine, household structure, and educational level, with accuracy losses around

3-5% if these variables were excluded. Marital status, nationality, and minors in care were uninformative or even noisy. In the living conditions model for anxiety, frequency of leaving home and residence attributes were the most informative with small impact, whereas features such as size or rooms had minimal. Also the living space arise as a noisy feature. In the economic model, most indicators, such as income and employment, were detrimental. In contrast, the perceived household situation during confinement was the only helpful factor, impacting only 1 or 2% loss in accuracy. Finally, in the health model, prior suicide attempts, recognised disability, need for self-care assistance, and general health condition had small influence. At the same time, COVID-related exposures and ongoing treatments were slightly noisy. These findings indicate that social restrictions, subjective perceptions of instability, and pre-existing vulnerabilities, rather than static demographic or economic markers, were the most consistent drivers of adverse mental health outcomes during confinement.

As an extra insight to facilitate a comparative analysis of the models across subsets and psychological conditions, a bar chart summarising these results is provided in Fig. 3.

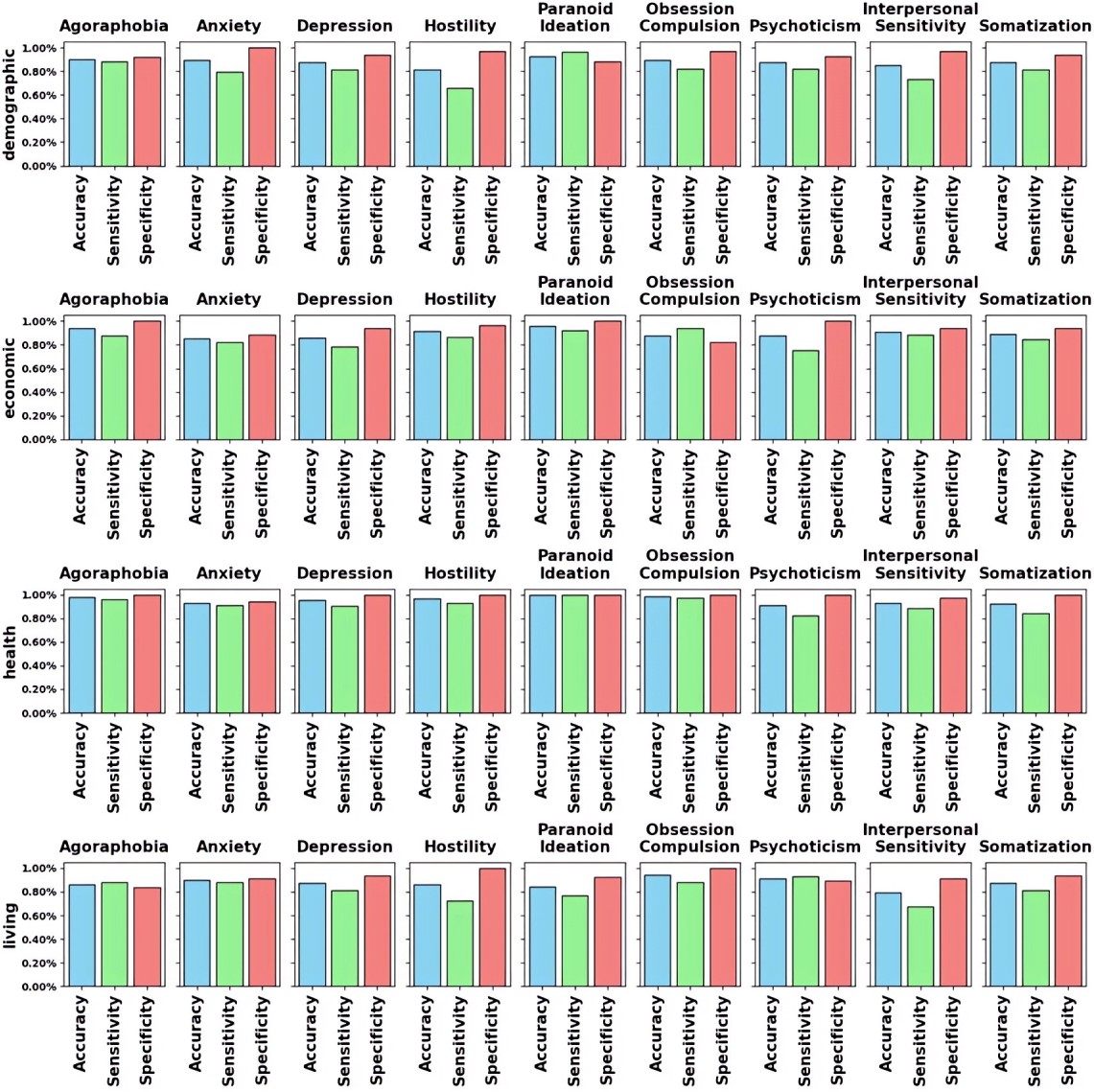

Figure 3: Comparison of different metrics in ML models' performance across subsets.

Fig. 3 provides an overview of the performance of the models across the three metrics that were evaluated. Ideally, the models would demonstrate high accuracy with minimal difference between specificity and sensitivity. In cases where one metric significantly outperforms accuracy, this must not be specificity, due to the problems this could cause, as previously discussed. Upon analysing the differences, only five cases show noteworthy deviations, such as anxiety within demographic features and psychoticism within economic characteristics. Instances where specificity exceeds accuracy are limited to three cases: paranoid ideation within demographic features, obsession-compulsion within economic characteristics, and paranoid ideation within living features. Overall, most models exhibit stable

performance with reliable metrics, with only three cases identified as potentially less trust-worthy.

To provide a more detailed analysis of the performance of the ML models, Fig. 4 and 5 were generated. Fig. 4 identifies the best-performing model for each use case across the four subsets.

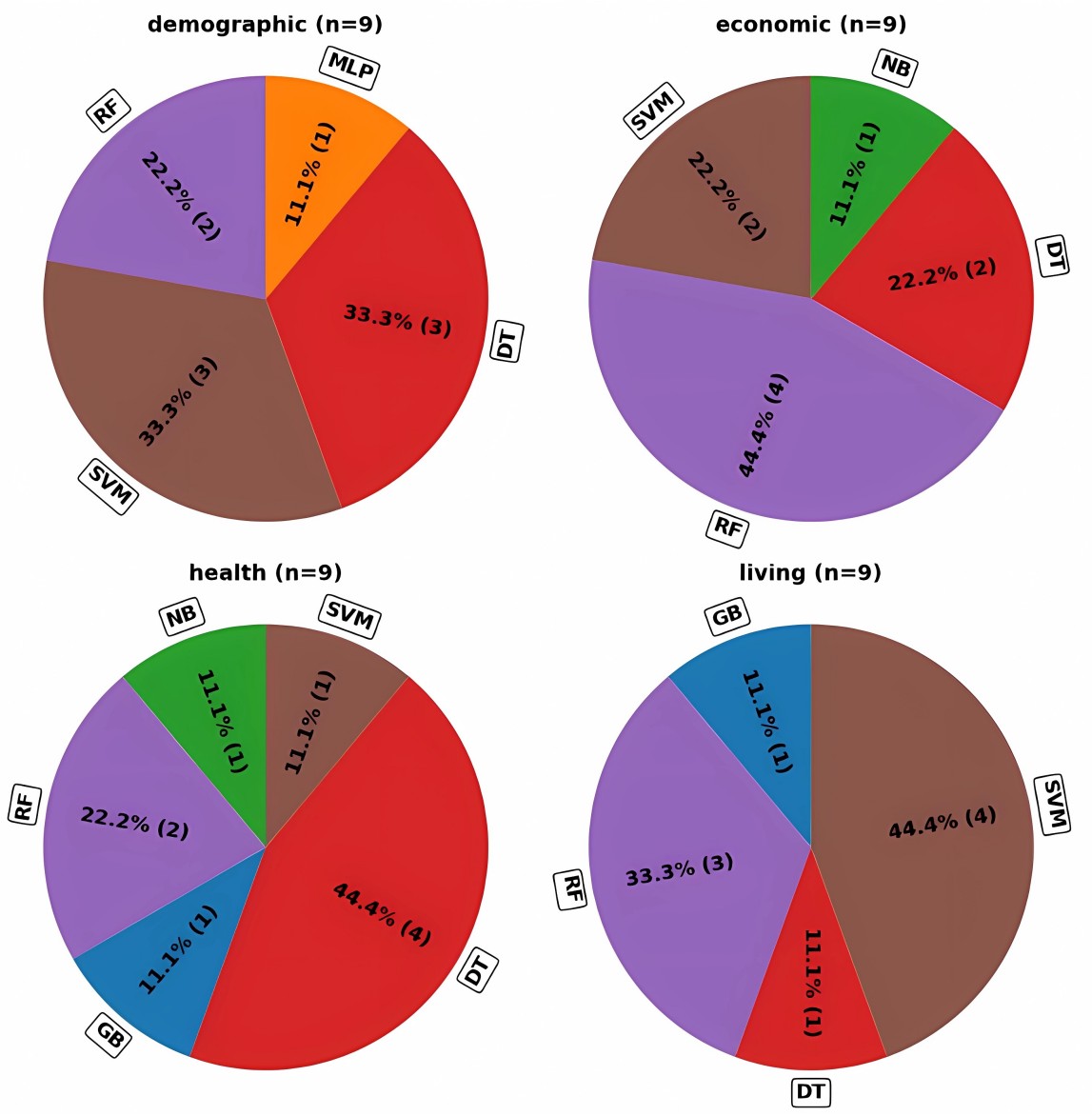

Figure 4: Distribution of the best ML models across the 4 subsets.

As illustrated in Figure 4, RF and SVM are consistently the best-performing models across all four subsets, indicating their adaptability to diverse data characteristics. DT also performs strongly, ranking as the top model in three subsets. Notably, LR is the only model

that never achieves the highest performance in any subset.

Fig. 5 presents a similar analysis, but focuses on identifying the best-performing model for each psychological condition.

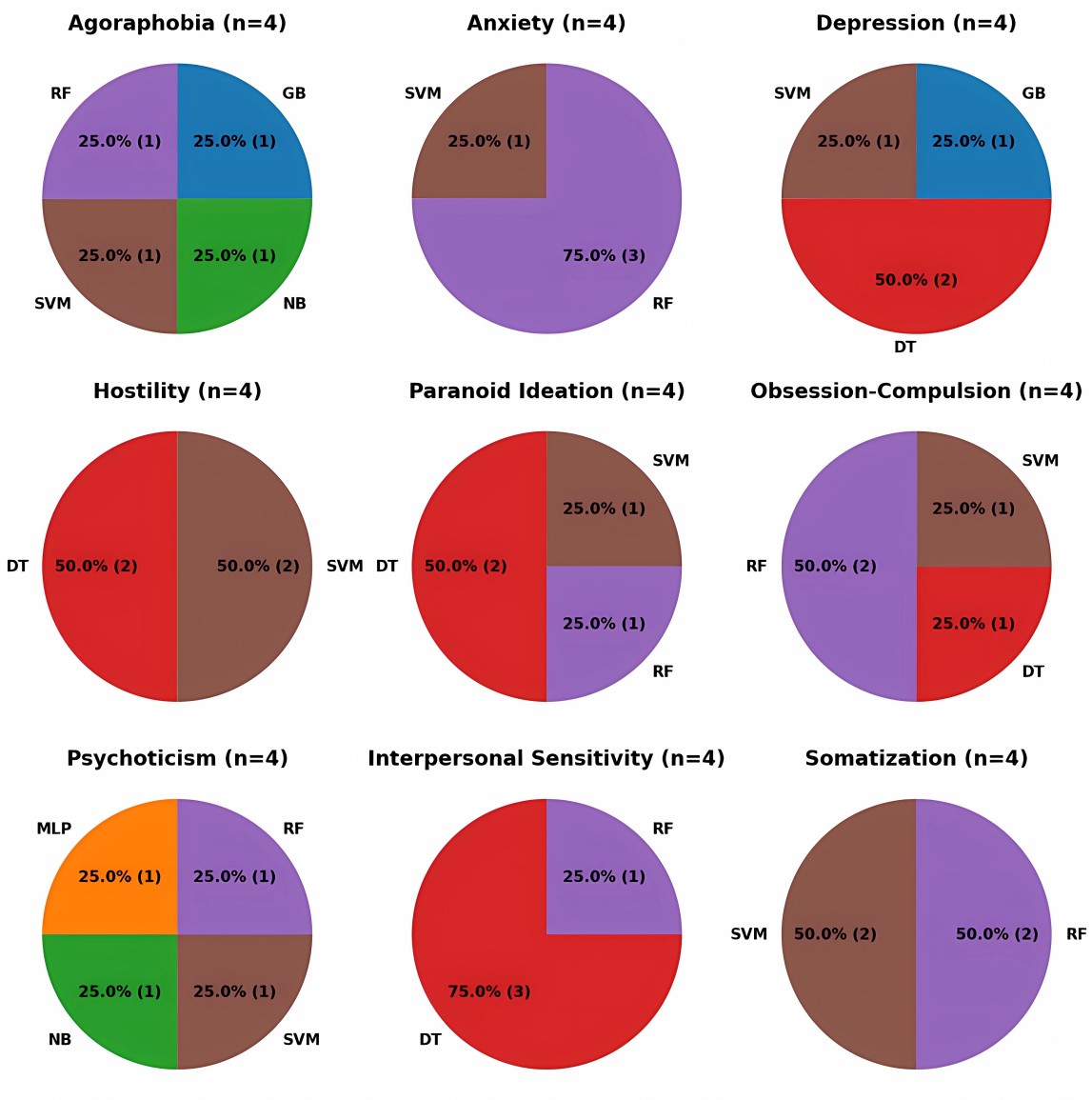

Figure 5: Distribution of the best ML models across the 9 psychological conditions.

Fig. 5 reveals that for the three psychological conditions, no ML model outperforms the others. In four of the psychological conditions, performance is distributed between two models. Finally, in two subsets, a single model achieves the best performance in 75% of the cases: RF for obsession-compulsion and DT for interpersonal sensitivity. Overall, RF and SVM perform best across six psychological conditions, while DT excels in five. In summary, RF emerges as the most suitable model, leading in 11 cases, followed by SVM and DT with

nine cases each.

For the second group of experiments, we have modelled a multiple linear regressor for each subset. Then, for each of the models, we have obtained the following values. The minimum p-value highlights the most statistically significant relationship within the model. $R^2$ indicates the proportion of variance explained by the predictors, and the adjusted $R^2$ refines this by accounting for the number of variables used. Finally, the Root Mean Squared Error (RMSE) measures the average prediction error, with lower values indicating better model accuracy. This information is compiled in Table 6.

Table 6: Regression Models Analysis

| Dataset | Min p-value | $R^2$ | Adjusted $R^2$ | RMSE |
|---|---|---|---|---|
| Demographic | 3.46E-09 | 0.0799 | 0.0695 | 0.9587 |
| Economic | 3.47E-06 | 0.0810 | 0.0725 | 0.9581 |
| Health | 1.03E-15 | 0.1700 | 0.1597 | 0.9106 |
| Living | 7.62E-09 | 0.0608 | 0.0521 | 0.9686 |

If we analyse information from Table 6, we can conclude the following. Across all models, at least one predictor shows a highly significant relationship with the outcome (very low min p-values), confirming that relevant variables exist in each subset. However, the overall explanatory power varies notably: the model using the health subset stands out with the highest $R^2$ (0.17) and lowest RMSE (0.91), indicating it explains a larger proportion of the variance and yields more accurate predictions compared to the other domains. The demographic, economic, and living subsets show moderate $R^2$ values (approximately 0.06–0.08) with similar RMSE values close to 0.96, suggesting these factors contribute meaningfully but explain less variance individually.

This study has several limitations. First, the absence of pre-lockdown baseline measures prevents estimation of person change due to the confinement. Second, the online snowball strategy introduced potential biases underrepresenting older adults and individuals with limited internet access. Third, the cross-sectional, single-wave design hinders causal inference and precludes trajectory analyses. Fourth, outcomes are self-reported without clinical verification. Finally, the binarisation of outcomes has no external normative validation.

## 5 Conclusions and future works

The primary aim of this work is to introduce a dataset connecting features from four different domains, demographic characteristics, living environment, economic status, and health impacts, with nine psychological conditions. This dataset is unique not only because of the variety of features collected, but also because of the specific context in which the data were gathered: during the strict quarantine measures in Spain during the COVID-19 outbreak in 2020. This context provides valuable insights into how these factors may have affected psychological well-being during an unprecedented global crisis.

To evaluate the usefulness of this dataset and as a complementary contribution to the work, we divided it into four subsets according to the nature of the features and then trained various ML models to assess their performance. The results demonstrate that the ML classifiers perform well in terms of accuracy across all subsets. However, challenges were observed in

terms of sensitivity and specificity values, highlighting the need for improved identification of positive cases. In terms of model performance, RF emerged as the top performer, followed by SVM and DT. However, the calibration curves for the diagnosis of agoraphobia show that the best-performing models are not the most reliable. Then, the regression analysis shows that while all domains include significant predictors, the health model explains the largest share of variance with better predictive accuracy. In contrast, the demographic, economic, and living models have lower explanatory power, highlighting the need for further model refinement for prediction use cases. Summarising, this paper introduces the LOCKED dataset and demonstrates its potential use through machine learning benchmarks. While diagnosis performance is promising, the primary value lies in the richness and diversity of the data, which we hope will serve the broader research community.

An important limitation of this dataset is the absence of baseline psychological measures prior to the lockdown. Due to the sudden onset of the COVID-19 confinement in Spain, it was not feasible to anticipate the situation and collect mental health data in advance. While our survey design includes retrospective items comparing pre- and during-lockdown conditions for variables such as income, employment, living situation, and social interaction, the SA-45 symptoms were assessed only during the lockdown. Therefore, it is not possible to directly measure psychological change attributable to the confinement, and caution should be exercised when making causal interpretations.

Looking ahead, future work will focus on a deeper exploration of which specific features within each subset have the most significant impact on predicting the nine psychological conditions. Identifying these influential features could provide valuable insights for mental health professionals, enabling more targeted interventions. Another complementary analysis is the obtaining of all the calibration curves to decide which are the most reliable models, while balancing the trade-off between accuracy and stability. Further analysis could also explore other ML techniques, such as ensemble methods or deep learning models, to improve prediction accuracy and address the sensitivity and specificity issues identified in this study. Understanding the dynamic relationships between these features and psychological health could also open up possibilities for future research into mental health diagnostics, personalised treatment strategies, and prevention for vulnerable populations.

Another important point relates to mitigating the limitations of the snowball sampling strategy, as older adults and individuals from less digitally connected regions may be underrepresented in the dataset. Future studies could consider applying reweighting or post-stratification techniques to better align the sample distribution with population demographics. Stratified analyses may also help assess whether the observed patterns hold across different age groups or levels of digital access. In addition, combining online surveys with targeted offline recruitment would help capture individuals who are less digitally connected, thereby improving representativeness and strengthening external validity.

## Broader Impact Statement

This work provides a unique dataset designed to improve our understanding of the mental health impacts of the COVID-19 lockdown in Spain, offering a rich combination of psychological assessments and socio-demographic data. By providing an open-access resource,

this research supports the further exploration of how personal features, living conditions, economic factors, general health, and mental health interact during crises, potentially informing public health policies and targeted interventions. On a positive note, this dataset enables the development of predictive models for mental health outcomes, providing tools that can identify vulnerable populations early on. These insights could be vital for designing equitable mental health interventions during future public health emergencies.

However, we acknowledge potential risks. Using Machine Learning models to predict mental health outcomes could raise concerns about data privacy, stigmatisation of individuals based on predictions, and overgeneralization if the models are applied without accounting for nuanced individual circumstances. These risks underscore the importance of ethical considerations and responsible application of findings to prevent unintended harm. Overall, this study emphasises the potential of integrating data-driven approaches to address complex societal challenges, while also urging caution in deploying such technologies to ensure their benefits are maximised equitably.

## Acknowledgments and Disclosure of Funding

No funding was received during this research.

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

## Appendix . Personal questionnaire

1. Please indicate the gender you identify with.
2. Date of birth (to select your year, click on the arrow next to the date and then scroll down the sidebar).
3. Indicate your marital status.
4. Please specify your place of birth.
5. What is the current status of your documentation?
6. Do you have European nationality?
7. What is the highest level of education you have completed?
8. Please provide details of your studies, if applicable.
9. Please indicate your usual living arrangements during the 12 months BEFORE the Lockdown.
10. Please indicate your usual living arrangements during the lockdown.
11. Please indicate the number of MINORS in your care during the quarantine.
12. Please indicate the number of dependents over 18 years old in your care (including elderly individuals and people with disabilities) during the quarantine.
13. Please indicate how many people you usually had face-to-face contact with on a normal day BEFORE the quarantine (including at home, work, and socially).
14. Please indicate how many people you usually had face-to-face contact with on a normal day, during the quarantine (including at home, work, and socially).
15. Please indicate your Postal Code during the lockdown.
16. Please specify the type of space you were living in during the lockdown.
17. Please indicate if you own your home or if you are renting.
18. How many usable square meters (that you can walk on) did your residence have during the quarantine?
19. Would you say your living space during the quarantine had adequate ventilation?
20. Would you say your living space during the quarantine had sufficient natural light?
21. Please indicate if these elements were present in your living space during the Quarantine.
22. Considering your place of residence during the lockdown, please indicate the number of rooms (not counting the bathroom and kitchen).
23. Considering your place of residence during the lockdown, please indicate the number of people you lived with.
24. Considering your place of residence during the lockdown, please indicate the number of people in your bedroom, excluding yourself.
25. Employment status BEFORE (in the 12 months before the start of the quarantine or most of the time).
26. Please provide details of the type of job during that period, if applicable.
27. What was the duration of your workday BEFORE (in the 12 months before the start of the quarantine or most of that time)?
28. What was your employment status during the quarantine?
29. What do you consider could be your employment or academic status AFTER the crisis caused by the coronavirus (in the 12 months following the end of the quarantine or most of that time)?

30. In the 12 months BEFORE the lockdown, what was the main occupation of the PERSON who contributed the most economic support to the household?

31. What was the approximate level of regular MONTHLY net income in your household (unit where expenses are shared: individual, couple, family) BEFORE the quarantine (in the 12 months before the start of the quarantine or most of that time)?

32. What was the approximate level of regular MONTHLY net income in your household (unit where expenses are shared: individual, couple, family) during the quarantine?

33. During the confinement in your home, did you experience any significant changes in your financial situation?

34. Suppose you (and your spouse or partner) convert all your funds in current and/or savings accounts, stock market investments, bonds, real estate, and sell your house, vehicles, and all your valuable items into money. Then, suppose you use the money from all these transactions to pay your mortgage and other debts, loans, debts, and credit cards. Would you still have money left after paying all your debts, or would you still owe money (a rough estimate is sufficient)?

35. Please indicate if you currently have any of the following health conditions.

36. If you have ever received a psychiatric diagnosis, please indicate which one.

37. Are you currently receiving psychiatric treatment or medication?

38. Are you currently receiving psychological treatment or therapy?

39. If you have consumed substances weekly in the last 6 months, please specify which ones.

40. Please indicate if you need help with daily self-care tasks such as shopping, household chores, bathing, grooming, cooking, managing money, etc.

41. Please indicate, if applicable, the degree of disability according to your certificate [a]42. Please indicate if you have ever attempted suicide.

43. Please indicate if you have been diagnosed with a coronavirus infection.

44. If you have had a coronavirus diagnosis, how would you rate the severity of the illness?

45. In this case, did you remain isolated inside your home (without leaving a room and without company during the duration of symptoms and 15 more days)?

46. If any member of the family unit has been diagnosed with a coronavirus infection, please indicate who it is (check more than one option if applicable).

47. If so, please assess the severity of the disease (consider the most severe case if there are multiple cases).

48. Has any member of your family diagnosed with a coronavirus infection lived in your home during their illness?

49. How many times a week did you leave the house during the quarantine?

50. For what reasons did you leave the house during the quarantine?

51. Do you consider that the measures taken to prevent the pandemic's progression are adequate and fair?

## Appendix . SA-45 Test

1. The idea that another person can control your thoughts.
2. Believing that most of your problems are someone else's fault.

3. Feeling scared in open spaces or on the street.
4. Hearing voices that other people do not hear.
5. The idea that most people cannot be trusted.
6. Feeling sudden and irrational fear.
7. Outbursts of anger or rage that you cannot control.
8. Fear of going out alone.
9. Feeling lonely.
10. Feeling sad.
11. Losing interest in things.
12. Feeling nervous or very anxious.
13. Believing that others are aware of your thoughts.
14. Feeling that others do not understand or listen to you.
15. Having the impression that people are unfriendly or that you are disliked.
16. Having to do things very slowly to be sure you are doing them right.
17. Feeling inferior to others.
18. Muscle pain.
19. The feeling that others are watching or talking about you.
20. Having to check everything you do repeatedly.
21. Having difficulty making decisions.
22. Feeling afraid to travel by bus, subway, or train.
23. Feeling hot or cold suddenly.
24. Having to avoid certain places or situations because they scare you.
25. Mind going blank.
26. Numbness or tingling in any part of your body.
27. Feeling hopeless about the future.
28. Having difficulty concentrating.
29. Feeling weak in any part of your body.
30. Feeling worried, tense, or agitated.
31. Heaviness in arms or legs.
32. Feeling uncomfortable when people look at you or talk about you.
33. Having thoughts that are not yours.
34. Feeling the urge to hit, hurt, or harm someone.
35. Feeling like breaking something.
36. Feeling very shy around other people.
37. Feeling scared or anxious in crowded places (like a cinema or supermarket).
38. Panic or terror attacks.
39. Having frequent arguments.
40. Feeling that others do not adequately recognise your achievements.
41. Feeling restless or uneasy.
42. The feeling of being useless or worthless.
43. Shouting or throwing things.
44. The impression that people would try to take advantage of you if they could.
45. The idea that you should be punished for your sins.

**Appendix . Distribution of Categorical Variables**

| Variable | Distribution | Entropy |
|---|---|---|
| Age | 26-35: 41% / 36-45: 30% / >46: 11% / 16-25: 10% / Unknown: 7% | 2 |
| Has anyone in your family diagnosed with coronavirus been with you during their illness? | Not applicable: 59% / No: 37% / Yes: 4% | 1.166 |
| Financial Sufficiency and Situation during COVID | They had enough money to support themselves: 84% / They had to borrow money from a friend or family member to support themselves: 9% / They had to request financial aid to support themselves: 5% / They did not have and could not request enough money to support themselves: 1% / They had to request a bank loan to support themselves: 1% | 0.873 |
| You would say that your place of residence during the quarantine has | A lot of natural light: 57% / Medium natural light: 35% / Little natural light: 8% | 1.284 |
| You would say that your place of residence during the quarantine has | A lot of ventilation: 64% / Medium ventilation: 33% / Little ventilation: 3% | 1.092 |
| If yes, please rate the severity of the illness (consider the most severe case if there were several). | Not applicable: 86% / Moderate: 6% / Mild: 5% / Severe: 2% / Death due to infection: 2% | 0.873 |
| If you have or had a diagnosis of coronavirus, how would you rate the severity of the illness? | Not applicable: 87% / Mild: 8% / Moderate: 4% / Severe: 2% | 0.765 |
| In this case, did you remain isolated inside your home (without leaving a room and without company during the duration of the symptoms and 15 additional days)? | Not applicable: 89% / Only partially: 7% / Yes: 4% | 0.604 |

| Variable | Distribution | Entropy |
|---|---|---|
| In the 12 months BEFORE the confinement, what was the main occupation of the PERSON who contributed the most income to the household? | Senior executives, owners of large businesses, professionals: 22% / Other: 17% / Junior professionals, artists, supervisors: 15% / Technicians, semi-professionals, owners of medium-sized businesses: 12% / Operators, semi-skilled workers: 9% / Unskilled workers: 7% / Retired or pensioner: 6% / Owners of very small businesses, skilled manual labourers, artisans, farmers: 6% / Sales, owners of small businesses: 5% / Day labourers, minor service jobs: 1% | 3.044 |
| Please indicate your marital status. | Married or cohabiting: 36% / In a relationship: 32% / Single: 28% / Divorced: 2% / Separated: 1% / Widowed: 1% | 1.817 |
| Please indicate whether you own your home or rent it. | Private rental: 46% / Private ownership (family): 22% / Private ownership (yours): 22% / No property or rental contract (living with friends or relatives, etc.): 8% / Not applicable: 2% / Public housing rental (state-provided): 1% | 1.947 |
| Have you been diagnosed with a coronavirus infection? | No: 79% / I may have had or am having the illness but was not tested: 20% / Yes, I had the infection and recovered: 1% / Yes, currently infected: 0% | 0.799 |
| Please indicate whether you need help with daily and self-care tasks such as grocery shopping, housework, washing and grooming, cooking, managing money, etc. | I do not need help: 96% / I need help: 4% | 0.242 |
| If applicable, please indicate the percentage of disability recognised in your certificate (0 if none). | 0: 97% / 50–95: 1% / 25–49: 1% / 5–24: 1% | 0.242 |
| Please indicate how many people you had face-to-face contact with on a typical day BEFORE the quarantine (including household, work, and social contacts). | 11 to 20 people: 34% / 1 to 10 people: 30% / 21 to 30 people: 16% / More than 50 people: 10% / 31 to 50 people: 9% / None: 1% | 2.185 |

| Variable | Distribution | Entropy |
|---|---|---|
| Please indicate how many people you usually had face-to-face contact with on a typical day during the quarantine (including household, work, and social contacts). | 1 to 10 people: 82% / None: 14% / 11 to 20 people: 3% / 21 to 30 people: 1% / 31 to 50 people: 1% | 0.917 |
| Please indicate the type of dwelling you lived in during the confinement. | Apartment: 83% / Semi-detached house: 8% / Detached house: 6% / Country house: 3% / Sheltered housing: 0% | 0.910 |
| Have you ever attempted suicide in the past? | No: 91% / Yes: 9% | 0.436 |
| Please indicate your usual living arrangement during confinement. | Partner: 46% / Own family: 20% / Family of origin: 14% / Alone: 12% / Friends: 5% / Roommates: 3% | 2.112 |
| Please indicate your usual living arrangement during the 12 months BEFORE confinement. | Partner: 42% / Own family: 21% / Alone: 13% / Family of origin: 11% / Roommates: 7% / Friends: 6% | 2.243 |
| Are you currently receiving psychiatric treatment or medication? | No: 89% / Yes: 11% | 0.500 |
| Are you currently receiving psychological treatment or attending therapy? | No: 82% / Yes: 18% | 0.680 |
| Employment status BEFORE (during the 12 months before the start of the quarantine or most of that time). | Employed (employee): 66% / Self-employed: 12% / Student/internship: 7% / Unemployed: 5% / Informal work (no contract): 3% / Retired: 2% / Inactive: 2% / Protected employment: 2% / Temporary disability (sick leave): 1% / Domestic worker: 1% | 1.871 |

| Variable | Distribution | Entropy |
|---|---|---|
| Suppose you (and your spouse or partner) convert into cash all your checking/savings accounts, stock market investments, bonds, real estate, and sell your house, vehicles, and valuables. Then, suppose you use all that money to pay your mortgage, loans, debts, and credit cards. Would you have money left after paying all your debts, or would you still owe money? (An approximate estimate is sufficient.) | You would have money left: 60% / You would still owe money: 27% / Debts would equal assets: 13% / Exact parity between debts and assets: 0% | 1.335 |
| Thinking about your place of residence during the confinement, please indicate the number of rooms (excluding bathroom and kitchen.) | 3: 33% / 4: 25% / 2: 17% / 5: 10% / 1: 7% / 6: 3% / 7: 1% / 8: 1% / >10: 1% / 0: 1% / 10: 0% / 9: 0% | 2.547 |
| Thinking about your place of residence during the confinement, please indicate the number of people you lived/live with. | 1–2: 83% / 0: 12% / 3–4: 4% / >10: 1% / 7–8: 0% / 9–10: 0% | 0.842 |
| What do you consider your likely employment or academic situation after the crisis caused by the coronavirus (during the 12 months following the end of confinement or for most of that time)? | On-site work as employee: 40% / Remote work as employee: 13% / Unemployed without benefits: 11% / Unemployed with benefits: 7% / On-site self-employed work: 6% / Studying from home: 5% / Flexible work schedule: 5% / Remote self-employed work: 4% / Temporary layoff (ERTE): 4% / Retired: 3% / Domestic worker: 1% / Other: 1% / Permanent layoff (ERE): 0% | 2.796 |
| What was the approximate level of regular net monthly income in your household (unit where expenses are shared: individual, couple, family...) before the quarantine (during the 12 months before the start of the quarantine or most of that time)? | €1000–1499: 23% / €1500–1999: 18% / €2000–2499: 15% / €2500–2999: 13% / €3000–3499: 9% / €500–999: 9% / €3500–3999: 6% / €4000–4499: 5% / Up to €499: 2% | 2.9924 |

| Variable | Distribution | Entropy |
|---|---|---|
| What was the approximate level of regular net monthly income in your household (unit where expenses are shared: individual, couple, family...) during the quarantine? | €1000–1499: 28% / €1500–1999: 14% / €500–999: 14% / €2000–2499: 12% / €2500–2999: 9% / Up to €499: 7% / €3000–3499: 7% / €3500–3999: 4% / €4000–4499: 3% / €4500–5000: 1% / €0: 0% | 2.929 |
| What is the current status of your documentation? | Valid passport: 94% / Residence visa/permit: 4% / Undocumented: 2% / Stay visa/permit: 0% | 0.383 |
| What is the highest level of education you have completed? | University studies: 39% / Postgraduate studies: 27% / Vocational training: 20% / Secondary education: 10% / Primary education: 2% / Incomplete primary education: 0% | 1.949 |
| What was your employment situation during the quarantine? | Remote work as employee: 33% / Temporary layoff (ERTE): 16% / Unemployed without benefits: 11% / On-site work as employee: 9% / Unemployed with benefits: 7% / Studying from home: 6% / Self-employed or business owner suspended due to COVID-19: 5% / Remote self-employed work: 4% / Retired: 3% / Flexible work schedule: 2% / Other: 2% / Domestic worker: 1% / On-site self-employed work: 1% / Permanent layoff (ERE): 0% | 3.038 |
| How many times per week did you go outside during the quarantine? | About every 10 days: 31% / On average 1 or 2 times a week: 28% / Other: 17% / Every day: 13% / I did not go out: 10% | 2.187 |
| How many usable square meters (the area you can walk on) did/does your place of residence have during the quarantine? | 76–90 m²: 22% / 46–60 m²: 17% / 61–75 m²: 16% / 91–105 m²: 11% / 30–45 m²: 10% / 106–120 m²: 8% / More than 180 m²: 7% / 121–150 m²: 4% / 151–180 m²: 3% / Less than 30 m²: 2% | 3.089 |
| What was the duration of your workday before the quarantine (in the 12 months before or most of that time)? | Full-time (approximately 40 hours): 71% / Part-time (approximately 20 hours): 17% / Not applicable: 7% / Less than 20 hours: 5% | 1.270 |
| Do you have European nationality? | European: 98% / Non-European: 2% 0.141 | |

## Appendix .  Distribution of Numerical Variables

| Variable | Mean | STD |
|---|---|---|
| Please indicate the number of minors under your care during the quarantine. | 0.3 | 0.7 |
| Please indicate the number of dependent persons over 18 under your care (including elderly or functionally diverse persons) during the quarantine. | 0.1 | 0.4 |
| Thinking about your place of residence DURING the confinement, please indicate the number of people in your same bedroom, excluding yourself | 0.7 | 0.7 |
| The idea that you should be punished for your sins | 0.2 | 0.7 |
| Change in cohabitation structure | 0.2 | 0.4 |
| Change in number of social relationships | 0.7 | 0.4 |
| Window | 0.8 | 0.4 |
| Private balcony | 0.0 | 0.0 |
| Private terrace | 0.3 | 0.5 |
| Private garden | 0.1 | 0.3 |
| Private outdoor land | 0.1 | 0.3 |
| Change in work format | 0.6 | 0.5 |
| Change in income | 0.5 | 0.5 |
| Health condition | 0.5 | 0.5 |
| Psychiatric diagnosis | 0.3 | 0.5 |
| Frequent substance use | 0.8 | 0.4 |
| Family member with COVID | 0.0 | 0.2 |

## Appendix . Distribution of initial features among the four subsets

Demographic characteristics:

- Gender

- Age

- Nationality

- Marital Status

- Current Documentation Status

- Education Level

- Family Structure before COVID

- Family Structure during COVID

- Change in Family Structure

- Number of Minors in Care during COVID

- Adults in Care Over 18

Living environment:

- Type of Living Space during Confinement

- Property Ownership

- Square Meters of Living Space

- Light during Quarantine

- Ventilation during Quarantine

- Number of Rooms

- Number of Cohabitants

- Number of Cohabitants in the Same Room

- Outings during Quarantine

- Assessment of COVID Measures

Economic status:

- Employment Status Before Quarantine

- Employment Status during Quarantine

- Employment Status After Quarantine

- Change in Employment Status

- Working Hours Before Quarantine

- Occupation of the Person with the Highest Economic Contribution

- Net Monthly Income Before COVID

- Net Monthly Income during COVID

- Change in Net Monthly Income

- Financial Sufficiency and Situation during COVID

- Financial Sufficiency and Situation After COVID

Health impacts:

- Health Condition and Specific Needs

- Disability Degree

- Suicide Attempt

- COVID Diagnosis

- COVID Severity

- Isolation during COVID-19 diagnosis

- Family Member Diagnosed with COVID

- Family Member COVID Severity

- Co-living with a Diagnosed Family Member

