# OpenReview forum: "LOCKED: A Dataset of Sociodemographic, Economic, Health and Living Features to Assess Mental Health Impact of the Spanish Lockdown during COVID-19"
_DMLR — Accepted by DMLR_

### Review · Reviewer_jApP · 2025-09-20

**Recommendation:** 3
**Confidence:** 2

**Summary Of Contributions:**

The paper introduces LOCKED, an openly available dataset collected in Spain during the strict COVID-19 lockdown, combining sociodemographic, economic, living and health variables with SA-45 assessments across nine dimensions. The revision clarifies that the primary contribution is the dataset, with ML models used only as illustrative benchmarks; the manuscript separates dataset creation (Section 3) from ML analyses (Section 4) and expands the conclusions accordingly. It now also includes sensitivity/specificity tables and a linear regression analysis on continuous SA-45 totals.

**Strengths:**

his submission makes a timely and substantive contribution by releasing an openly available, well-documented dataset on mental health during Spain’s strict COVID-19 lockdown, integrating rich sociodemographic, economic, living-condition and health variables with a standardized instrument (SA-45) across nine dimensions. The revised manuscript clearly frames the work as a dataset paper and cleanly separates the dataset description from illustrative ML benchmarks, improving clarity and purpose. The additions—more explicit curation and preprocessing details, comprehensive descriptive statistics, and a continuous-outcome regression analysis alongside classification baselines—enhance methodological quality and reproducibility. The provision of code and data meaningfully lowers barriers to reuse, supporting transparent validation and extension.

Relative to prior work on crisis-time mental-health surveys, LOCKED stands out for its breadth of variables, the use of a recognized clinical measure, and a transparent pipeline that will be valuable to multiple communities, including mental-health informatics, public health, and machine learning. The paper is clearly written and organized, making it straightforward to understand the dataset’s scope, limitations, and intended use. Ethical and social implications are addressed with appropriate caution—e.g., acknowledging convenience-sampling biases and the absence of pre-lockdown baselines, and avoiding diagnostic claims—thereby encouraging responsible use while maximizing the dataset’s potential for societal benefit.

**Audience:**

Yes

**Broader Impact Concerns:**

The manuscript adequately addresses broader impact issues related to mental health assessments during crises. However, potential ethical considerations around the risk of stigmatization or misclassification due to simplified binary mental health assessments should be explicitly discussed.

**Claims And Evidence:**

Claims about predictive capability are now supported by accuracy plus sensitivity/specificity across nine conditions and by a linear regression on continuous SA-45 totals (with modest R² except in the health subset). Framing these as baselines/illustrative use cases is now consistent with the paper’s aim.

**Datasets And Benchmarks:**

Openly available (Zenodo link) and code on GitHub; please provide a stable DOI and a datasheet/data card for longevity and reuse.

**Extended Submissions:**

The manuscript does not appear to be an extended version of previously published work. It meets originality criteria for submission.

**Limitations:**

• No pre-lockdown baseline: prevents estimating within-person change attributable to confinement.

• Convenience/snowball sampling: likely self-selection and coverage biases; sample not reweighted to census → limited representativeness.

• Cross-sectional single wave: hinders causal inference and trajectory analysis.

• Self-report measurement without clinical validation: potential recall/social-desirability bias; SA-45 is non-diagnostic.

• Heuristic binarization of outcomes: thresholds lack external validation/calibration, affecting class prevalence and performance metrics.

**Requested Changes:**

1. SA-45 thresholds: briefly justify the binarization rule and add a short sensitivity check (e.g., alternative cutoffs) or provide calibration curves for at least one condition.
2. Interpretability/ablation: add a concise feature-importance/SHAP plot and, if possible, an ablation by domains (demo / living / economic / health) for one representative task.

**Strengths And Weaknesses:**

Strengths:
- Clarity & scope improved: clear positioning as a dataset paper with ML as usage notes; explicit sectioning (Sec. 3 vs Sec. 4).
- Curation & descriptives: added detailed preprocessing/curation (NaNs strategy, outlier handling) and appendices with distributions/entropy for categorical and means/SDs for continuous variables; geographic spread across Spanish regions is reported.
- Recruitment transparency: recruitment channel, window (2–9 May 2020), eligibility and sampling bias caveats are now described.
- Benchmarks strengthened: beyond accuracy, the paper reports sensitivity/specificity and adds linear regression (R², adj-R², RMSE) on SA-45 totals; test splits and grid-search procedures are documented; code repo provided.
- Limitations acknowledged: no pre-lockdown SA-45 baseline is explicitly recognized.

Weaknesses
- Thresholding for SA-45: binarization remains heuristic (mean+1SD) without external validation; consider adding calibration/robustness or linking to norms if available.
- Interpretability/ablation: still no SHAP/permutation importances or ablation by variable blocks; desirable to guide users on which domains drive predictions.
- Ethics statement: consent via Google Forms is noted, but no IRB/ethics approval ID is reported; advisable to clarify the oversight pathway expected for a public dataset.

---

### Review · Reviewer_SEm2 · 2025-09-22

**Recommendation:** 4
**Confidence:** 2

**Summary Of Contributions:**

The paper introduces LOCKED, a publicly available dataset collected during Spain’s strict COVID-19 lockdown. It combines sociodemographic, economic, health, and living condition data with psychological assessments (SA-45) of nine mental health conditions. Machine learning benchmarks demonstrate high predictive accuracy (>80%), highlighting the dataset’s value for analyzing lockdown-related mental health impacts

**Strengths:**

The submission makes a significant contribution by presenting the LOCKED dataset, a novel and publicly accessible resource capturing the mental health impacts of Spain’s strict COVID-19 lockdown. Its uniqueness lies in the integration of four domains—demographic, economic, health, and living features—with validated psychological assessments (SA-45). This multidimensional design offers researchers an opportunity to study the complex interplay between social conditions and mental health outcomes in unprecedented circumstances.

**Audience:**

Yes

**Claims And Evidence:**

yes

**Datasets And Benchmarks:**

yes

**Extended Submissions:**

no

**Requested Changes:**

The dataset relies on online snowball sampling, which underrepresents older adults and those with limited digital access. A more explicit discussion of how this affects external validity, along with possible mitigation strategies (e.g., reweighting, stratification), is essential.

Current benchmarks emphasize accuracy, but given the clinical importance of minimizing false negatives, metrics such as F1, ROC–AUC, precision, and recall should be reported. This would provide a more balanced view of predictive performance, particularly for imbalanced classes.

**Strengths And Weaknesses:**

pros:
1. The LOCKED dataset is unique in capturing the mental health impact of Spain’s highly restrictive COVID-19 lockdown, combining psychological assessments (SA-45) with sociodemographic, economic, health, and living variables. Unlike many related datasets, LOCKED is openly accessible, making it a valuable contribution for reproducibility and further research.
2. The dataset integrates 41 curated features spanning four domains (demographics, living environment, economic status, health), enabling multidimensional analysis of psychological outcomes.

cons:
1. Regression models yielded low R2 values (0.06–0.17), which indicates limited explanatory power for continuous outcomes despite significant predictors.
2. While classification accuracy is reported, more nuanced metrics (e.g., F1, AUC) could strengthen the evaluation, especially given imbalanced classes and the clinical importance of minimizing false negatives.